# Selection and Identification of an ssDNA Aptamer for Fibroblast Activation Protein

**DOI:** 10.3390/molecules28041682

**Published:** 2023-02-09

**Authors:** Xiaomin Zhang, Ge Yang, Yi Zhao, Xuyan Dai, Wenjing Liu, Feng Qu, Yuanyu Huang

**Affiliations:** 1Key Laboratory of Molecular Medicine and Biotherapy, Key Laboratory of Medical Molecule Science and Pharmaceutics Engineering, School of Life Science, Advanced Research Institute of Multidisciplinary Science, Beijing Institute of Technology, Beijing 100081, China; 2CAMS Key Laboratory of Antiviral Drug Research, Beijing Key Laboratory of Antimicrobial Agents, NHC Key Laboratory of Biotechnology of Antibiotics, Institute of Medicinal Biotechnology, Chinese Academy of Medical Sciences and Peking Union Medical College, Beijing 100050, China; 3Economic College, Hunan Agricultural University, Changsha 410128, China; 4Beijing Key Laboratory of Drug Resistance Tuberculosis Research, Beijing Tuberculosis and Thoracic Tumor Research Institute, Beijing Chest Hospital, Capital Medical University, Beijing 101125, China

**Keywords:** FAP, aptamer, CE, SELEX, cancer-associated fibroblasts

## Abstract

As a type II transmembrane serine protease, fibroblast activation protein (FAP) is specifically expressed on the surface of fibroblasts associated with a variety of epithelial-derived malignancies such as pancreatic cancer, breast cancer, and colon cancer. It participates in the processes of tumorigenesis, progression, and immunosuppression. FAP constitutes an important target for tumor treatment; however, the current studies on FAP are mainly related to structural characteristics, enzymatic properties, and biological functions, and aptamers of FAP have not been investigated. In this work, by using recombinant human FAP as the target, five candidate aptamers, which are AptFAP-A1, AptFAP-A2, AptFAP-A3, AptFAP-A4, and AptFAP-A5, were selected by capillary electrophoresis–systematic evolution of ligands by exponential enrichment (CE-SELEX), and their secondary structures were predicted to be mainly stem-loop. Moreover, the CE-laser-induced fluorescence (LIF) method was used to determine the equilibrium dissociation constant *K_D_* values between the FAP protein and candidate aptamers, and the *K_D_* value was in the low molar range. Finally, Cy5-labeled aptamers were co-incubated with human pancreatic cancer-associated fibroblasts highly expressing FAP protein, and confocal microscopy imaging showed that aptamer AptFAP-A4 had the highest affinities with the cells. The FAP aptamers screened in this study provide a promising direction for the development of rapid tumor diagnosis and targeted therapy.

## 1. Introduction

Fibroblast activation protein (FAP) is a type of serine protein hydrolase that was originally identified as an antigen in fibroblasts cultured with monoclonal antibody F19 in 1986 [1,2]. FAP can be expressed in the stromal fibroblasts of more than 90% of epithelial tumors, including pancreatic, colorectal, gastric, and lung cancers, but it is generally not expressed in cancer cells, normal fibroblasts, or normal tissues [3,4,5,6]. CAFs are an important component of the tumor microenvironment (TME) and are strongly associated with the development, invasion, and metastasis of tumors [7,8,9,10]. FAP is considered a protein specifically expressed by CAFs, and by using FAP as a target for CAFs, the development of FAP monoclonal antibodies [11,12] and small molecule inhibitors [13,14,15,16,17] of FAP enzyme activity has been used in tumor therapy as well as diagnostic imaging. However, the clinical application of FAP has numerous limitations due to the complex preparation process and high production costs of the antibodies and inhibitors involved, as well as the immunogenicity of the antibodies.

Aptamers, known as “chemical antibodies”, are novel biological recognition molecules with high specificity and high affinity [18,19]. Compared with antibodies, aptamers have a wide range of targets, including metal ions [20,21], growth factors [22], peptides [23], proteins [24], glycans [25], and even cells [26]. In addition, aptamers have other advantages, such as simple preparation, low production costs, easy chemical modification, and non-immunogenicity. Until now, different types of aptamers have been widely developed for disease treatment and diagnosis [25,27]. For example, in the therapeutic aspects, aptamers modified with GC sequences were affixed with Adriamycin to specifically target multiple myeloma cells and inhibit tumor growth in multiple myeloma [28]; in addition, aptamer-engineered NK cells generated by anchoring the aptamer to the surface of NK cells can specifically bind CD30-expressing lymphoma cells without genetic alteration [29]. In the field of aptamer diagnosis, the electrochemical aptasensor, prepared by immobilizing aptamers on the surface of gold electrodes and achieving signal amplification through ST-ALP, can directly detect chronic granulocytic leukemia K562 cells [30].

Typically, aptamers, which are single-stranded oligonucleotides, are screened from a library of nucleic acid molecules using an in vitro screening technique (systematic evolution of ligands by exponential enrichment, SELEX). Depending on the different separation approaches of nucleic acid–target complexes, SELEX methodology can be divided into nitrocellulose membrane filtration-SELEX [31,32], microfluidic-SELEX [33,34,35], surface plasmon resonance-SELEX [36], magnetic beads-SELEX [37,38], capillary electrophoresis-SELEX (CE-SELEX) [39,40], etc. Among them, CE-SELEX is one of the most efficient screening approaches for aptamers [39,41,42]. In the CE-SELEX approach, first, one mixes the nucleic acid library with the target and then performs capillary electrophoresis. During electrophoresis, capillaries are used as separated channels, and the complexes and free nucleic acids are separated by different charge-to-mass ratios under a high-voltage electric field. Subsequently, the sequences of the complexes are amplified and purified to generate secondary libraries. Finally, aptamers are obtained by repeating a simple 1-4-round screening. The CE-based aptamer screening technique can achieve the separation, analysis, and collection of complexes simultaneously and has the advantages of high efficiency, rapidity, microvolume, and multimodality [39,43].

In the present study, FAP-specific aptamers were successfully screened by using a capillary electrophoresis-based SELEX strategy. A mixture containing the ssDNA nucleic acid library with the FAP protein was first co-incubated. Subsequently, the target–nucleic acid complexes were separated from the mixture via capillary electrophoresis, and five FAP candidate aptamer sequences were obtained by repeating four rounds of screening. Later on, the affinities of the candidate aptamers were characterized, and the specific binding of the candidate aptamers to the target cells was verified at the cellular level. The FAP-specific aptamers screened in this paper have the potential to be utilized as novel diagnostic or therapeutic tools for the detection and treatment of tumors.

## 2. Results

### 2.1. Aptamer Screening by CE-SELEX

To screen aptamers that can specifically bind FAP proteins, ssDNA was co-incubated with the target protein and subsequently screened for aptamers using the CE-SELEX method. In order to obtain the optimal analysis conditions, the injection volume, sample concentration, voltage used for separation, temperature, and time were optimized. Via CE-LIF analysis, the migration time of ssDNA was determined to be about 7.5 min (Figure 1A), and the peaks were sharp and symmetrical. After mixing the ssDNA with the target protein into the sample, a new peak appeared at 3.0–4.0 min, and the area of the new peak increased gradually, while the peak area of the free nucleic acid library decreased significantly. This indicates that during high-voltage electrophoresis, the FAP protein binds some of the ssDNA in the nucleic acid library and forms a new, stable complex. Then, the complexes, collected at the exit end of the capillary, were used as amplification templates for the secondary library for PCR amplification, and the PCR products were recovered via agarose gel electrophoresis for gel recovery (Figure 1B–D). The PCR products were used as secondary libraries for screening and amplification in the next round. Furthermore, we performed high-throughput sequencing on the PCR products obtained from the last round of screening. Due to the heterogeneity of CE-SELEX, more than 50,000 sequences were obtained after high-throughput sequencing. The five most enriched sequences were selected as candidate aptamers (AptFAP-A1~AptFAP-A5), and their performance was verified (Table 1).

### 2.2. Predicting Structure and Motif of Candidate Aptamers

The secondary structures of the five candidate aptamers were simulated using the NUPACK and M-fold applications (Figure 2). AptFAP-A2, AptFAP-A3, and AptFAP-A5 are large circular structures with 1 to 3 short arms. In contrast, AptFAP-A1 and AptFAP-A4 are circular structures with long arms. Among the five candidate aptamers, AptFAP-A4 had the lowest ∆G value and the largest Tm value, which indicates that it is more easily folded into a stable secondary structure. The ∆G values of AptFAP-A2 and AptFAP-A3 were higher than other candidate aptamers and had the lowest Tm values, indicating that both of them were structurally unstable (Table 1). Moreover, the 3dRNA/DNA tertiary structure prediction method was used to simulate the three-dimensional spatial structure of five candidate aptamers.

Using the web-based application WeLogo3 to generate a sequence logo (Figure 3A) of the candidate aptamer, the distribution of bases in the sequence and the characteristics of the concordant sequence could be visually depicted. The accumulation of residues at every position reflects the consistency of residues at that position. For example, there is only one capital letter C and G in column 2 and column 36, respectively, which indicates that the sequence at that position is conserved. Conversely, the shorter the height of a specific letter in the stack, the smaller the frequency of the nucleotide at that position, and, therefore, the sequence letters in that column are relatively confusing and have high entropy values. Subsequently, the MEME Suite was used to find and analyze the motifs of the candidate aptamers (Figure 3B). Setting the number of mined structural domains (motifs) to three, the program was run to obtain three different motifs, which are represented by red, blue, and green squares. There are three different motifs in the AptFAP-A2 sequence, with the “ATTAGTCT” sequence starting at position +6, “TGCGTACC” at position +24, and “ACCAYT” at position +15 (Figure 3C), while both AptFAP-A1 and AptFAP-A4 sequences have only one motif.

### 2.3. Molecular Docking and Kinetic Analysis

Molecular docking simulations of the candidate aptamers and FAP protein were performed using the PyMOL software, and the optimal docking model was selected based on the binding free energy. There were two types of interactions observed between the aptamer AptFAP-A1 and the FAP protein complexes (Figure 4A), including eight hydrogen bonds and four salt bridges. Some of the nucleotides formed hydrogen bonds or salt bridges by binding to amino acids at eight sites in the α/β hydrolase structural domain of the FAP protein. More specifically, the eight sites are Asn500, Glu615, Arg690, Gln721, Lys55, Lys564, Arg605, and Lys616. Three types of interactions were observed among the aptamer AptFAP-A2 and the FAP protein complexes (Figure 4B), including ten hydrogen bonds, five salt bridges, and four hydrophobic interactions. However, the interaction sites did not belong to either the α/β hydrolase structural domain or the catalytic structural domain. Two types of interactions, including six hydrogen bonds and six salt bridges, were observed between the aptamer AptFAP-A3 and FAP protein complexes (Figure 4C). To form hydrogen bonds or salt bridges, T63 and C62 were bound to Asp672 and Arg590, which were located in the structural domain of the α/β hydrolase in the B chain of the FAP protein, respectively, while A4 was bound to glycosylation site Asn227 in the B chain of the FAP protein. There were 3 types of interactions observed between the aptamer AptFAP-A4 and FAP protein complexes (Figure 4D), which included 11 hydrogen bonds, 3 salt bridges, and 1 hydrophobic interaction, in which T77 was bound to Asn49, Ser737, and Gly738 in the α/β hydrolase structural domain of the FAP protein to form hydrogen bonds, and Asn49 belonged to the glycosylation site of the FAP protein. Four types of interactions, including fourteen hydrogen bonds, five salt bridges, one hydrophobic interaction, and one pi-cation interaction, were observed among the aptamer AptFAP-A5 and FAP protein complexes (Figure 4E). Among them, there were 13 amino acids, such as Lys45, Asn49, and Gln721, which were located in the α/β hydrolase structural domain, interacting with the nucleotides in AptFAP-A5. 

Combined with the analysis results of the motifs of the candidate aptamers, it can be seen that, in the analysis results of AptFAP-A1~AptFAP-A4, the nucleotides interacting with the α/β hydrolase structural domains or glycosylation sites of the FAP proteins are non-conserved nucleotides. None of them belonged to the nucleotides in the motif. Conversely, in the analysis results of AptFAP-A5, there is only one nucleotide, T50, belonging to the motif to bind to the Lys753 and His750 of the FAP protein and form salt bridges and pi–cation interactions. The above results indicate that the conserved motifs in candidate aptamers are almost not bound to the important structural domains in FAP proteins. Moreover, the results of the *K_D_* value determination and colloidal gold analysis show that AptFAP-A2 has the lowest affinity for FAP proteins, which may be related to the presence of three conserved motifs in the sequence. Conversely, AptFAP-A1 and AptFAP-A4 are the two candidate aptamers with the highest level of binding to FAP proteins, and both of them contain only one conserved motif in their sequences. Therefore, we infer that the affinities of the candidate aptamers might be negatively correlated with their motif numbers. In addition, if candidate aptamers are involved in the interaction with the important structural domains in the target protein structure, such as the hydrolysis domain, catalytic region, and glycosylation site, the affinities and targeting levels may be higher.

### 2.4. Affinity Characterization of Candidate Aptamers

In order to determine the affinities of the candidate aptamers toward the target proteins, the equilibrium dissociation constants between the candidate aptamers and targets were determined using a CE-LIF detector (Figure 5A–E), and the magnitudes of the *K_D_* values of the five candidate aptamers (AptFAP-A1~AptFAP-A5) were calculated according to a specific formula (Figure 5F). The quantity *K_D_* was used to reflect the degree of affinity between the aptamers and targets; the larger the value of *K_D_*, the smaller the affinity, and vice versa. The research results showed that the *K_D_* values of AptFAP-A2, AptFAP-A3, and AptFAP-A5 were all larger than 1 μmol/L, while the *K_D_* values of AptFAP-A1 and AptFAP-A4 were comparable, which were both about 0.5 μmol/L.

A nanogold colorimetric assay was used to verify the specificity of the candidate aptamers toward the target proteins. The principle is that aptamers are adsorbed on AuNPs when there are no targets, and the aggregation effect of nanogold is prevented; therefore, the solution appears red. When the targets are added, the target binds to the aptamer to make the aptamer separate from the surface of the AuNPs, and the state of the nanogold changes from dispersed to aggregated, which results in the solution becoming blue. In the present experiment, candidate aptamer sequences of 20, 50, 100, 200, and 1000 nmol/L were co-incubated with FAP proteins. With the increase in the candidate aptamer concentration, the absorption peaks of the AuNPs gradually migrated from 520 nm to 620 nm (Figure 6A–E), while the color of the AuNPs and the position of the absorption peaks remained unchanged after the control proteins were bound to the candidate aptamer (Figure 6F). In particular, when 100 nmol/L AptFAP-A1 and 100 nmol/L AptFAP-A4 were added, the color of the AuNP solution changed from burgundy to blue, and the absorption peaks of the AuNPs all shifted to the right. The above results indicate that candidate aptamers AptFAP-A1 and AptFAP-A4 had good affinity and specificity, which corresponds to the *K_D_* value measured by the CE-LIF method. Moreover, the interactions between the candidate aptamers and other related proteins were assessed using CE-LIF. Five proteins were introduced for aptamer-specific evaluation, including CD26, a transmembrane serine protease similar to FAP; trypsin, a secreted serine protease; CD105, a transmembrane glycoprotein; and human thrombin (H-Thr) and human serum albumin (HSA), high-abundance interfering proteins in the blood. The experimental results reveal that the candidate aptamers have good specificity and no significant binding to the five proteins mentioned above (Appendix A).

### 2.5. Candidate Aptamers’ Affinity for Target Cells

In order to determine the binding affinities of the candidate aptamers for human pancreatic cancer-associated fibroblasts, five Cy5-labeled candidate aptamers, AptFAP-A1~AptFAP-A5, were diluted to a final concentration of 5 μmol/L. After that, human pancreatic cancer-associated fibroblasts digested from culture dishes were incubated with five candidate aptamers in a binding buffer, and, finally, the treated cells were imaged in real time using laser confocal microscopy. The results show that four of the candidate aptamers, AptFAP-A1, AptFAP-A2, AptFAP-A3, and AptFAP-A5, had a poor affinity for the target cells, and there was no obvious red fluorescence. In contrast, aptamer AptFAP-A4 was able to specifically bind to human pancreatic cancer-associated fibroblasts (Figure 7). Furthermore, to investigate the influence of FAP aptamers on cell growth, human pancreatic cancer-associated fibroblasts and Ramos cells were co-incubated with 1 μmol/L each of the five candidate aptamers for 24 h, and cell viability was subsequently detected and calculated using a CCK8 assay. The experimental results showed that candidate aptamers AptFAP-A1~AptFAP-A5 almost did not affect the growth of human pancreatic cancer-associated fibroblasts (Appendix A).

## 3. Materials and Methods

### 3.1. Reagents and Cell Lines

The ssDNA library (5′-AGCAGCACAGAGGTCAGATG-N40-CCTATGCGTGCTACCGTGAA-3′), primer1 (5′-AGCAGCACAGAGGTCAGATG-3′), primer2 (5′-TTCACGGTAGCACGCATAGG-3′), and aptamers were synthesized by Sangon Biotech Co., Ltd. (Shanghai, China). A capillary electrophoresis instrument with a fluorescence detector (Beckman P/ACE MDQ) was purchased from Beckman-Coulter Co., Ltd. (California, USA). Neutral-coated capillaries (75 μm inner diameter) were purchased from Handan Xinnuo Fiber Optic Chromatography Co., Ltd. (Handan, China), and 2×Taq Plus PCR Master Mix, Nucleic Acid Dye Gene Green, 50 bp DNA Ladder, and 5×TBE Buffer were purchased from Tiangen Biotech Co., Ltd. (Beijing, China). Recombinant Human FAP (carrier-free) was purchased from BioLegend Biotechnology Co., Ltd. (Beijing, China). Human pancreatic cancer-associated fibroblasts were purchased from Guangzhou Speed Research Biotechnology Co., Ltd. (Guangzhou, China). Cell Counting Kit-8 was purchased from YEASEN Biotech Co., Ltd. (Shanghai, China), and Fibroblast Medium Supplement was purchased from M&C Gene Technology Co., Ltd. (Beijing, China). The purity specifications of all chemical reagents were analytical reagents.

### 3.2. CE-SELEX Assays

The stock solution of the ssDNA nucleic acid library was first diluted to a certain concentration, and in order to fully denature the ssDNA nucleic acid library, it was incubated in a 95 °C water bath for 5 min, followed by slow cooling to room temperature. The treated ssDNA nucleic acid library was mixed and incubated with FAP protein in equal volumes. The capillaries need to be rinsed sequentially with 1.0 mol/L NaOH, water, and electrophoresis-running buffer before use. The mixture was separated, analyzed, and collected using CE at a pressure feed of 0.5 psi for 5 s, a separation voltage of 20 kV, and a separate temperature of 25 °C. PCR amplification was performed on the collected complexes, followed by electrophoresis and gel recovery of the PCR amplification products using 2% agarose gels; when it could generate sufficient amounts of secondary libraries, we proceeded to the next round of screening.

### 3.3. Cloning and Sequencing

By running PE300 mode in the Illumina MiSeq sequencing platform, high-throughput sequencing (HTS) was performed on PCR products obtained via CE-SELEX screening in the last round. Furthermore, deleting the primer dimers of the sequencing results first, the sequences were sorted from high to low frequency. Finally, candidate aptamer sequences were screened using frequency analysis to perform subsequent affinity characterization and specificity assays.

### 3.4. Structure and Motif Predictions

By using NUPACK and M-fold software to simulate the secondary structures of candidate aptamers, their thermodynamic parameters and the magnitude of their Tm values were further analyzed. Moreover, the 3D structures of five candidate aptamers were predicted by using the 3dRNA/DNA tertiary structure prediction method (http://biophy.hust.edu.cn/new/3dRNA (accessed on 21 November 2022). There was a difference in the binding sites of different candidate aptamers when bound to target proteins. To better elucidate the motifs of sequences and their roles in binding to target proteins, the online tools MEME (http://meme-suite.org/ (accessed on 23 November 2022) and WebLogo3 (http://weblogo.threeplusone.com/ (accessed on 23 November 2022) were used to predict and analyze the motifs of candidate aptamers without primers. The MEME suite is an online application that combines various functions such as motif discovery, motif enrichment, and motif comparison. The algorithm of MEME is based on the maximum expectation (EM) algorithm to identify motifs. Furthermore, WebLogo3 provides a graphical representation of residues appearing in every position, and the larger the character of a specific residue, the more frequently it appears in that position. 

### 3.5. Binding Mode Predictions

The amino acid sequences and 3D structures of the FAP proteins were obtained from the UniPort database (https://www.uniprot.org/ (accessed on 21 November 2022) and the RCSB database (https://www.pdbus.org/ (accessed on 21 November 2022), respectively. The HNADOCK server (http://hdock.phys.hust.edu.cn/ (accessed on 22 November 2022) was used to perform the docking of aptamers to FAP proteins, and then the Pymol software was used to analyze the results of molecular docking. Using the HNADOCK server (http://hdock.phys.hust.edu.cn/ (accessed on 22 November 2022) to dock aptamers to FAP proteins, the docking task was started after submitting the PDB files of the FAP protein (PDB ID: 1Z68) and aptamers to the web interface of the HDOCK server. The status page was refreshed every 10 s, and the docking process typically took 1 h. Then, Pymol molecular visualization software was used to present the predicted model and analyze the docking results.

### 3.6. Affinity Characterization

We determined the equilibrium dissociation constants between candidate aptamers and targets using a CE-laser-induced fluorescence (LIF) detector based on the nonequilibrium capillary electrophoresis of equilibrium mixtures (NECEEM) method [44,45,46] established by Krylov’s group. The *K_D_* values were calculated according to the following equations [47], where [P0] and [DNA] denote the concentration of FAP protein and ssDNA, respectively. In addition, A1, A2, and A3 denote the peak area of free ssDNA, the peak area of the dissociated region, and the peak area of the complex, respectively.
(1)$$K_{D}=\frac{\left[P0\right]\left(1+\frac{A1}{A2+A3}\right)-\left[\mathrm{DNA}\right]}{1+\left(A2+A3\right)/A1}$$

The affinities of the candidate aptamers were verified by using a nanogold colorimetric assay. First, gold nanoparticles (AuNPs) were prepared using the citrate reduction method. Next, 25 μL of AuNP solution and FAP target protein were added to a 96-well plate and incubated for 15 min at room temperature. Then, 25 μL of five candidate aptamer solutions with different concentration gradients were added and incubated for 20 min at room temperature. Finally, 10 μL of 0.9 mol/L NaCI solution was added and mixed well, and we let it react for 5 min. The UV absorption spectrum of the solution in the 96-well plate was detected using an enzyme standardization instrument.

### 3.7. Targeting of Aptamers to Cells

First, five candidate aptamer sequences were diluted to 5 μmol/L using PBS solution, which was 200 μL for each solution. Afterward, cells in the culture dish were digested with trypsin and counted. We let the cells be co-incubated with different concentrations of the aptamer sequences for 1 h at 37 °C, and then the cells were washed three times with PBS solution. After transferring it to confocal dishes, the cells were kept for 4 h to adhere to the wall. Finally, the cell membrane and nucleus were stained with Dio and Hoest33342, respectively, and washed 3 times with PBS. Then, a laser confocal microscope was used for imaging. In addition, to verify the safety of the candidate aptamers, a CCK8 kit was used to detect the influence of the candidate aptamers on cell proliferation. Selecting Ramos cells as the control group, cell suspensions (2 × 10^4^ cells/mL, 100 μL) were first inoculated into 96-well plates and then precultured. Subsequently, 1 μmol/L of each candidate aptamer was added to each well of the plate, and they continued to incubate for 24 h. Finally, adding CCK-8 solution to each well, an absorbance at 450 nm was measured using an enzyme marker to calculate cell viability.

## 4. Discussion

CAFs are the predominant stromal cells in the tumor microenvironment and are strongly associated with tumorigenesis, progression, immunosuppression, and drug resistance [7,48]. FAP is overexpressed in the stromal fibroblasts of most epithelial malignancies [49], and it can be used as a potential target for tumor diagnosis and treatment. Current methods used for FAP detection mainly include immunohistochemistry, Western Blot, liquid-phase mass spectrometry, and fluorescence detection methods [50,51,52,53]. However, these methods have limitations in various aspects, such as sensitivity, targeting, throughput, and interference resistance. Therefore, the development of rapid and efficient detection approaches for FAP is an important research direction. In addition, FAP as a target can achieve the effective inhibition of tumor growth. For example, Sostoa et al. [54] used a FAP-targeting bispecific T cell engager (FBiTE) to carry oncolytic adenovirus (OAd), which can not only enhance the lytic effect of the virus on tumor cells but also increase the cytotoxic effect mediated by FBiTE on FAP-expressing CAFs. As a result, it effectively enhanced anti-tumor activity. In another study, Wang et al. [55] developed a retroviral CAR construct specific to mouse FAP. The transduced muFAP-CAR mouse T cells were able to effectively target the tumor mesenchyme and enhance the anti-tumor response of endogenous CD8+ T cells. Nevertheless, in order to achieve the targeted recognition of FAP by T cells, a genetic modification of T cells is required, which not only leads to genetic mutations or off-target effects [56,57] but also increases production costs. This poses a challenge for immune-based therapies targeting FAP.

Aptamers have been successfully used as multifunctional tools in various fields, such as tumor diagnosis, targeted therapy, food detection, and analysis, as well as bioimaging. More specifically, as recognition elements, aptamers can be used not only to make different types of aptasensors [58,59,60,61] to detect and diagnose diseases but also to modify them to immune cells [29,62,63,64,65], which can impart targeting functions on immune cells.

In the present research, with recombinant human FAP protein as the target, five candidate aptamer sequences were obtained in four rounds of efficient and rapid screening using the CE-SELEX screening method. Afterward, the candidate aptamers were evaluated using structural simulations and thermodynamic parameters. Compared with other candidate aptamers, AptFAP-A4 has the lowest Gibbs free energy and the highest Tm value, which indicates that it is easier to fold to form a stable secondary structure, and it is less prone to unstranding. Subsequently, using the MEME Suite, the motif prediction and analysis of the candidate aptamer sequences without primers showed that both the AptFAP-A1 and AptFAP-A4 sequences had only one motif, while the AptFAP-A2 sequence contained three motifs. From the molecular docking results between the candidate aptamers and FAP proteins, it is clear that the forces between the nucleotides in the candidate aptamers and the amino acids in the FAP proteins involve hydrogen bonds, salt bridges, hydrophobic interactions, and pi–cation interactions. In particular, the binding site of AptFAP-A2, which contains three motifs, to the FAP protein does not belong to the hydrolytic enzyme structural domain or glycosylation site, while the binding of the other four candidate aptamers to the FAP protein involves the hydrolytic enzyme structural domain or glycosylation site. Therefore, this reveals that the motif in the aptamer sequence may affect the affinity of the aptamer for the target protein. It also shows that motif analysis and molecular dynamics simulations can help select the best aptamer sequence. The equilibrium dissociation constant *K_D_* values between the FAP protein and candidate aptamers were determined using the CE-LIF method in a range of 0.551–2.182 μmol/L, and the affinity and specificity were further verified via nanogold colorimetry. To evaluate the affinities of the candidate aptamers at the cellular level, the candidate aptamers were co-incubated with human pancreatic cancer-associated fibroblasts and subsequently imaged using laser confocal microscopy. The results showed that, compared with the other four candidate aptamers, aptamer AptFAP-A4 had a higher affinity for human pancreatic cancer-associated fibroblasts with high FAP protein expression at the cellular level.

FAP aptamers are a promising targeting tool for disease diagnosis and molecular targeting therapy. Aptasensor, prepared by using a FAP aptamer as a recognition element, is expected to achieve high sensitivity and high-throughput detection for samples. In addition, the modification of FAP aptamers on the surface of immune cells can reduce interference to the inherent anti-tumor biological functions of immune cells, and as a non-genetic approach, it also has the advantages of flexibility, controllability, and predictability. In subsequent work, for the purposes of constructing engineered T cells with dual-targeting of CAF and tumor cells, the FAP aptamer can be modified to CAR-T cells or co-modified with another aptamer onto T cells. T cells with dual-targeting functions are expected to reverse the immunosuppression induced by the tumor microenvironment and further enhance the efficacy of the adoptive cellular immunotherapy.

## 5. Conclusions

After performing four rounds of replicate screening using the CE-SELEX method, we selected five candidate aptamer sequences (AptFAP-A1~AptFAP-A5) from the ssDNA library, based on their enrichment levels and sequencing frequencies, to perform functional validation. Among them, AptFAP-A4 was the most thermodynamically stable aptamer (∆G = −7.29 kcal/mol) with the maximum Tm value (62.5 °C). The nucleotides involved in the binding of candidate aptamers AptFAP-A1~AptFAP-A4 to FAP proteins were not conserved motifs. Moreover, candidate aptamers AptFAP-A1 and AptFAP-A4 had lower *K_D_* values and stronger affinities. At the cellular level, AptFAP-A4 had the strongest affinity for the target cells. In conclusion, applications of the FAP aptamers screened in the current study are promising for the diagnosis and treatment of tumor diseases in future research work.

## Figures and Tables

**Figure 1 molecules-28-01682-f001:**
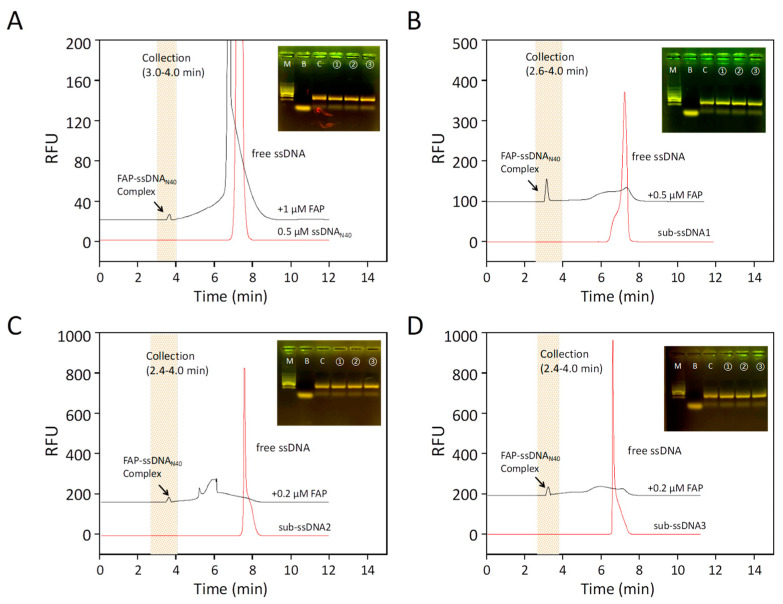
CE-SELEX for FAP aptamers. (**A**) The capillary electrophoresis chromatogram of the first round of screening. The red chromatogram shows the 0.5 μmol/L nucleic acid libraries (ssDNA_N40_) alone, and the black chromatogram shows the incubated mixture of 0.5 μmol/L ssDNA_N40_ and 1 μmol FAP. Arrows indicate the complex formed by the combination of FAP and ssDNA_N40_. The shaded part is the complex collection segment of 3.0–4.0 min. (**B**) The second round of screening. The red chromatographic line is the secondary library (sub-ssDNA1), amplified and purified after the first round of screening; the black chromatographic line is the mixture of sub-ssDNA1 incubated with 0.5 μmol/L FAP; the arrow indicates the complex formed by the combination of FAP and sub-ssDNA1. The collection interval was 2.6–4.0 min. (**C**,**D**) show the third and fourth rounds of screening. The red chromatographic lines are sub-ssDNA2 and sub-ssDNA3, and the black chromatographic lines are 0.2 μmol/L FAP incubated with sub-ssDNA2 or sub-ssDNA3. The collection interval was 2.6–4.0 min for both. Lanes of the gel electrophoresis pattern: M is marker; B is blank control; C is PCR control with the nucleic acid library as a template; and ①, ②, and ③ are parallel samples of PCR products.

**Figure 2 molecules-28-01682-f002:**
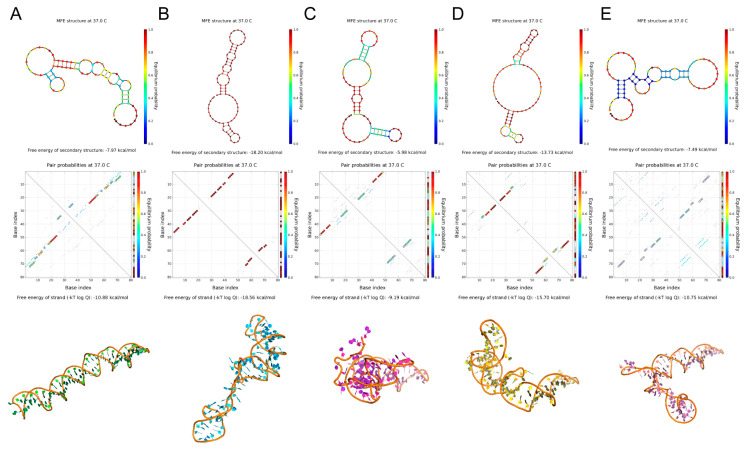
Secondary and tertiary structure prediction of candidate aptamers. (**A**–**E**) The secondary and tertiary structures of AptFAP-A1~AptFAP-A5 were predicted by the NUPACK software and the 3dRNA/DNA tertiary structure prediction method, respectively.

**Figure 3 molecules-28-01682-f003:**
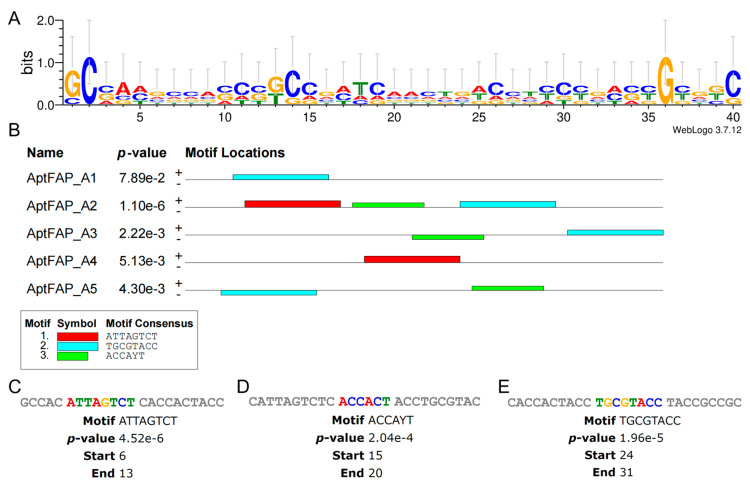
Motif prediction and analysis of candidate aptamers. (**A**) Sequence logo of candidate aptamers generated by the WeLogo3 program. (**B**) Three different motifs of candidate aptamers generated by the MEME Suite program. Each color square represents a motif. (**C**–**E**) Motif sequence and specific location in AptFAP-A2. The candidate aptamer sequences used for motif prediction and analysis do not contain forward primers (P1) or reverse primers (P2).

**Figure 4 molecules-28-01682-f004:**
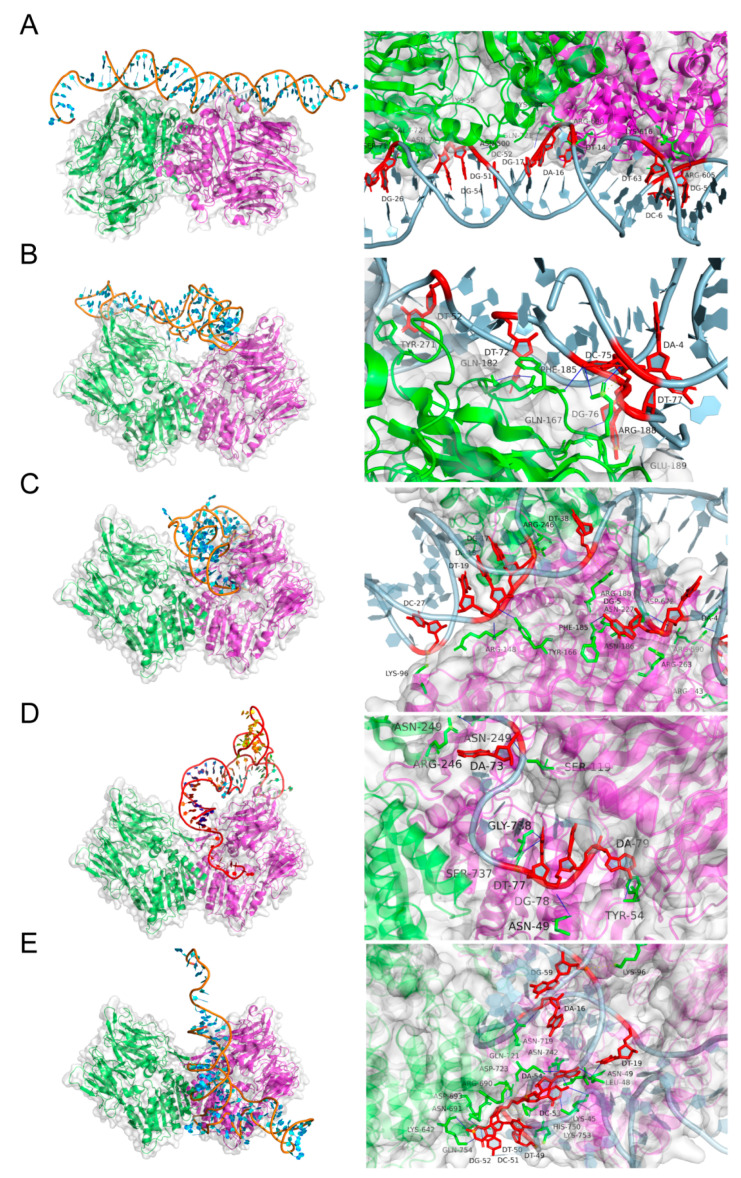
Three-dimensional structure and docking prediction. (**A**–**E**) Molecular docking models of AptFAP-A1~AptFAP-A5 with FAP proteins and details of the binding sites. The right figure shows the specific binding sites of the candidate aptamer and FAP protein, in which the amino acid residue is shown as stick models in green, the nucleotide residue is shown as stick models in red.

**Figure 5 molecules-28-01682-f005:**
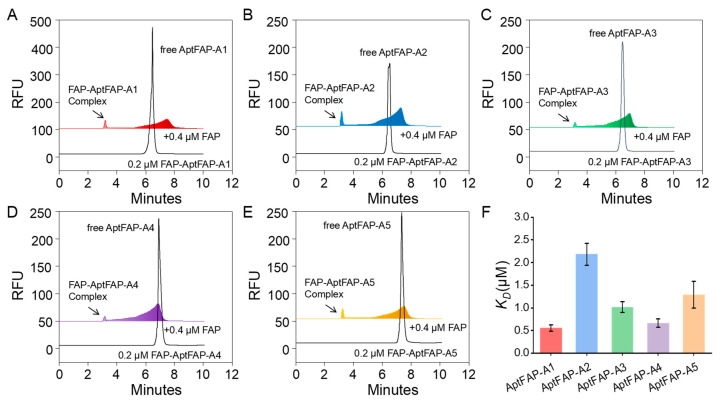
Determination of interactions and *K_D_* values between candidate aptamers and FAP proteins using the CE method. (**A**–**E**) Characterization of the affinities of candidate aptamers AptFAP-A1~AptFAP-A5; (**F**) calculation of *K_D_* values using the NECEEM method.

**Figure 6 molecules-28-01682-f006:**
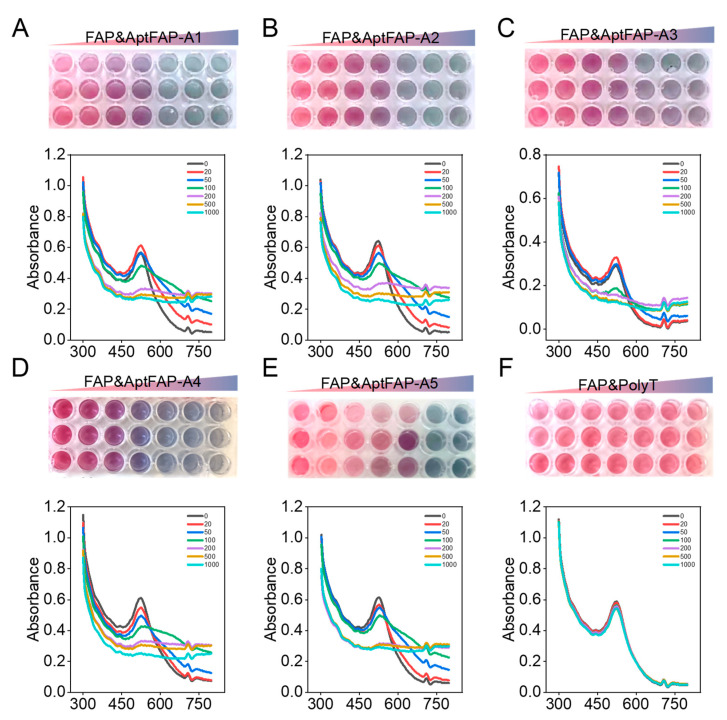
The affinities of candidate aptamers were verified using colloidal gold colorimetry. (**A**–**F**) UV absorption spectra of target proteins after co-incubation with candidate aptamers AptFAP-A1~AptFAP-A5 and the control group PolyT, respectively. With the increase in the candidate aptamer concentration, the absorption peak of the AuNP solution gradually shifted from 520 nm to 620 nm, and the color changed from burgundy to blue.

**Figure 7 molecules-28-01682-f007:**
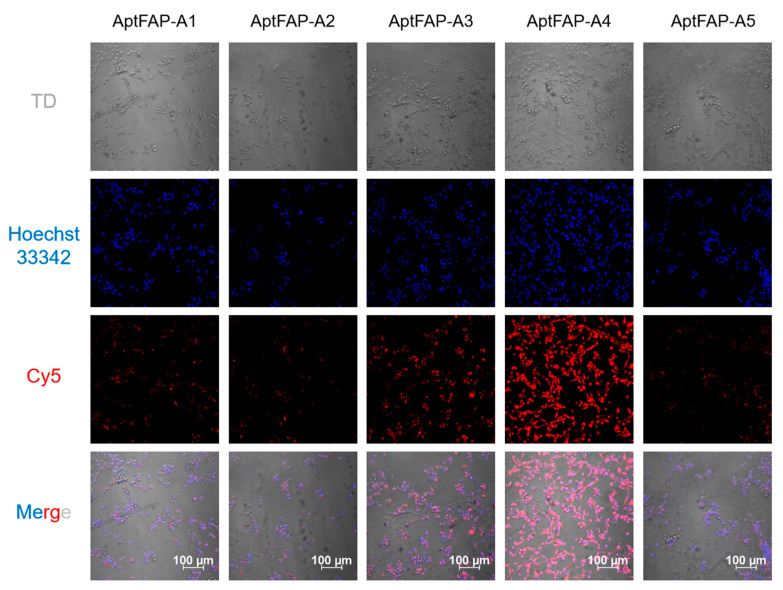
Binding of candidate aptamers to target cells. At the cellular level, AptFAP-A4 showed the highest affinity for target cells.

**Table 1 molecules-28-01682-t001:** Sequences and thermodynamic parameters of five candidate aptamers.

Name	Sequence (5′-3′)	∆G (kcal/mol)	∆H (kcal/mol)	∆S (cal/K·mol)	Tm (°C)	Seq-Frequency
AptFAP-A1	P1-GCGAAGCGTACCGGCTACCCAGTGACAGTCGCCGTGGGTC-P2	−2.45	−59.20	−182.9	50.3	347
AptFAP-A2	P1-GCCACATTAGTCTCACCACTACCTGCGTACCTACCGCCGC-P2	−1.78	−55.00	−171.5	47.3	311
AptFAP-A3	P1-GCCATCCCCCCGTCCGATGAGTGGTGCTCCTATGCGTTCC-P2	−1.42	−53.30	−167.2	45.4	59
AptFAP-A4	P1-CCGCAGGCAGCTGCCATTAGTCTCTATCCGTGACGGTATG-P2	−7.29	−95.90	−285.7	62.5	18
AptFAP-A5	P1-GCAGCTAAGCAGGCGGCTCACAAAACCATTCGCATGCGGC-P2	−3.07	−83.00	−257.7	48.9	17

Note: The primers that are invariant at both ends of the aptamers are P1: 5′-AGCAGCACAGAGGTCAGATG-3′ and P2: 5′-CCTATGCGTGCTACCGTGAA-3′.

## Data Availability

Not applicable.

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
