# Peer review of "Selection and Identification of an ssDNA Aptamer for Fibroblast Activation Protein"

_molecules, 2023, doi:10.3390/molecules28041682_

Round 1
Reviewer 1 Report
This manuscript reported that, by using CE-SELEX, five ssDNA aptamers (apts) binding to human FAP, were identified. The authors suggested the aptamers can be used as diagnostic or therapeutic tools for the detection and treatment of FAP-related tumours. My major concern is the authors only provide few in vitro data. Whether treatment of the FAP-apts will affect the level (e.g., by Western blotting in cells) and activity (e.g., inhibit the dipeptidyl peptidase and collagenase activities by biochemical assays) were not explored. Whether treatment of FAP-apts affect the growth of cells especially for the FAP- cells to show any other potential toxicity effects. In addition, it is better to perform pull down assays with mass spectrometry analysis to show whether FAP-apts also bind to other proteins.
Reviewer 2 Report
In this manuscript Zhang et. al, has focused on interesting study of targeting FAPS with aptamers, the approach used by authors is fresh and could open more avenues if the work continued in right direction. Authors have selected the 5 candidate aptamers by using CE-SELEX approach, further authors predicted the aptamer structures and studied the MD simulations. KD values of selected aptamers were determined using colorimetric assay, in addition to confocal microscopy to analyze the affinity of selected aptamers with human pancreatic cancer-associated fibroblast.
It would be more convincing of authors could also perform in vivo studies with mouse model and show the real time use to aptamers.
Authors have claimed about CE-SELEX to be the most efficient amongst the other techniques, the supporting reference would strengthen the claim of the authors in introduction.
The description of Aptamer screening results authors could add the optimized values or optimized conditions for injection volume, sample concentration, voltage used for separation, temperature.
The Figure1 needs the detailed descriptive legend mentioned the which figure belonged to what aptamer. The gel electrophoresis figures need to be labelled, which lane has what samples. What are the graphs indicating in Panel A, B, C and D needs to be mentioned in detailed, the description of figure legend in detail will help the readers to have better understanding about the respective result.
I would suggest authors to include more details in result section while describing Figure 1 and Table 1 such as what is the optimal range of Tm values for the selection of enlisted aptamers. Also include the initial number of aptamers, i. e. out of how many screened aptamers, the enlisted 5 were selected stating the rationale of selection. The arrangement of the aptamer sequence also required to be arranged in an alternate way, the current arrangement makes the differentiation among primers difficult.
The figure legends of all figures are needed to be described in detail for example in Figure 3, the aptamer motif used in structure prediction should be mentioned in legend. Also, authors could also discuss the energy values of the predicted structures and add the table in supplementary information. The validation of predicted structures could be the value addition to the structure prediction.
The results could be rearranged having MD simulation after structure prediction, that will maintain the continuous flow of the results.
Authors could also add the time duration for how long the MD simulations were run along with the details of environment.
Discission needs to be rewritten with detailed results discussion. The inference drawn from the results should be compared with state of art research and other findings. Although authors have discussed other FAP targeting approaches but it would be more helpful for readers if authors could provide direct comparison present study and other studies implementing aptamers targeting FAP.
Conclusion needs to be added separately.
Round 2
Reviewer 1 Report
The authors did not address my concerns. Show the apts can inhibit the activity of FAP "is a must", and shall not difficult to do (e.g., https://www.ncbi.nlm.nih.gov/pmc/articles/PMC3871272/).
The experimental design to address my concern "Point 2" is problematic. Whether those cells expressing FAP, if not what is the point to do the experiment. If FAP is expressing in the cells, why inhibit their FAP without any effect on their growth. FAP specific siRNA shall be used in this experiment as control and the level and activity of FAP shall be examined by Western blotting and in vitro assay.
I was dumbstruck with their experimental design of experiment to address my concern "Point 3". Their Apts did not bind to those five proteins does not mean they will not bind to others.
Reviewer 2 Report
The authors have included the suggested changes.
Author Response
Dear Reviewers,
Thanks very much for your review and the recognition of our research. It will be the greatest encouragement for young researchers.
Kind regards.